# T2-high Asthma, Classified by Sputum mRNA Expression of *IL4*, *IL5*, and *IL13*, is Characterized by Eosinophilia and Severe Phenotype

**DOI:** 10.3390/life11020092

**Published:** 2021-01-27

**Authors:** Matija Rijavec, Tomaž Krumpestar, Sabina Škrgat, Izidor Kern, Peter Korošec

**Affiliations:** 1University Clinic of Respiratory and Allergic Diseases Golnik, 4204 Golnik, Slovenia; tomaz.krumpestar@gmail.com (T.K.); sabina.skrgat@gmail.com (S.Š); izidor.kern@klinika-golnik.si (I.K.); peter.korosec@klinika-golnik.si (P.K.); 2Biotechnical Faculty, University of Ljubljana, 1000 Ljubljana, Slovenia; 3Department of Pulmonary Diseases, University Medical Centre Ljubljana, 1000 Ljubljana, Slovenia; 4Faculty of Medicine, University of Ljubljana, 1000 Ljubljana, Slovenia

**Keywords:** asthma, endotype, T2-high asthma, gene expression, IL-4, IL-5, IL-13, T2 gene mean, eosinophilic, severe asthma, biological treatment

## Abstract

Asthma is a common chronic disease, with different underlying inflammatory mechanisms. Identification of asthma endotypes, which reflect a variable response to different treatments, is important for more precise asthma management. T2 asthma is characterized by airway inflammation driven by T2 cytokines including interleukins IL-4, IL-5, and IL-13. This study aimed to determine whether induced sputum samples can be used for gene expression profiling of T2-high asthma classified by *IL4*, *IL5,* and *IL13* expression. Induced sputum samples were obtained from 44 subjects, among them 36 asthmatic patients and eight controls, and mRNA expression levels of *IL4*, *IL5*, and *IL13* were quantified by RT-qPCR. Overall, gene expression levels of *IL4*, *IL5*, and *IL13* were significantly increased in asthmatic patients’ samples compared to controls and there was a high positive correlation between expressions of all three genes. T2 gene mean was calculated by combining the expression levels of all three genes (*IL4*, *IL5*, and *IL13*) and according to T2 gene mean expression in controls, we set a T2-high/T2-low cutoff value. Twenty-four (67%) asthmatic patients had T2-high endotype and those patients had significantly higher eosinophil blood and sputum counts. Furthermore, T2-high endotype was characterized as a more severe, difficult-to-treat asthma, and often uncontrolled despite the use of inhaled and/or oral corticosteroids. Therefore, the majority of those patients (15 [63%] of 24) needed adjunct biological therapy to control their asthma symptoms/exacerbations. In conclusion, we found that interleukins *IL4*, *IL5,* and *IL13* transcripts could be effectively detected in sputum from asthmatic patients. Implementation of T2 gene mean can be used as sputum molecular biomarker to categorize patients into T2-high endotype, characterized by eosinophilia and severe, difficult-to-treat asthma, and often with a need for biological treatment.

## 1. Introduction

Asthma is a common, highly heterogeneous, chronic disease characterized by the inflammation and narrowing of the airways [1]. Being one of the most common chronic diseases, affecting more than 300 million people worldwide, represents a serious challenge for the public health system [1,2]. Asthma is a highly treatable disorder with several therapies available, however many patients with severe, uncontrolled asthma suffer from ongoing symptoms and exacerbations despite maximal optimized therapy [1]. Even though severe asthma affects a minority of patients, the majority of medical resources are directed towards those patients [1,2]. Large asthma clinical heterogeneity and high variability in treatment response might be explained by different asthma endotypes. Identification and understanding of the molecular mechanisms of different asthma endotypes, that reflect a highly variable response to different treatments, will lead to more precise asthma management and better outcomes in patients [2,3]. In recent years, several biomarkers of clinical response for asthma endotypes have been proposed. Gene expression profiling in induced sputum cells has been shown as a robust method for classification of asthma into different endotypes, especially T2-high/T2-low, each having specific inflammatory, remodeling, and treatment response characteristics [4,5,6,7,8,9,10]. T2 asthma is characterized by airway inflammation driven by T2 cytokines including interleukins IL-4, IL-5, and IL-13 [5].

This study aimed to test the feasibility of induced sputum gene expression profiling in a clinical setting and to determine whether induced sputum samples can be used for gene expression profiling and if T2-high endotype can be classified based on *IL4*, *IL5,* and *IL13* profiling. Furthermore, clinical characterization and clinical differences between T2-high and T2-low asthma were also analyzed.

## 2. Results

### 2.1. Higher IL4, IL5, and IL13 mRNA Expression in Induced Sputum Cells of Asthmatic Patients

We have found significantly increased gene expression levels of *IL4*, *IL5*, and *IL13* in induced sputum cells from 36 asthmatic patients in comparison to eight control subjects without chronic diseases (Figure 1A). Furthermore, a high positive correlation between *IL4*, *IL5,* and *IL13* expression was also observed. Specifically, as shown by the Spearman’s rank correlation analysis *IL4* and *IL5* (*r_s_* = 0.89; *p* < 0.0001), *IL4* and *IL13* (*r_s_* = 0.90; *p* < 0.0001), and *IL5* and *IL13* (*r_s_* = 0.91; *p* < 0.0001) mRNA expressions were all positively correlated (Figure 1B).

### 2.2. T2-High Asthma is Characterized by Eosinophilia and Severe, Difficult-to-Treat Asthma

T2 gene mean was calculated by combining the mRNA expression levels of *IL4*, *IL5,* and *IL13.* T2 gene mean significantly inversely correlated with FEV1 (%) preBD (*r_s_* = −0.36, *p* = 0.029), and positively correlated with blood (*r_s_* = 0.42, *p* = 0.021) and sputum eosinophils (*r_s_* = −0.63, *p* < 0.001), as well as the need for biological therapy (*r_s_* = 0.53, *p* < 0.001) to control their asthma symptoms. Additionally, T2 gene mean also marginally correlated with the need for oral corticosteroids (*r_s_* = 0.32, *p* = 0.054) to control asthma symptoms (Figure 2).

Based on the expression of T2 gene mean in controls, the cut-off value to discriminate between T2-high and T2-low was set at −0.162. Accordingly, 24 out of 36 (67%) of asthmatic patients were classified as having T2-high asthma (Table 1). Patients with T2-high asthma had significantly higher eosinophil blood (*p* = 0.035) and sputum (*p* = 0.017) counts, lower FEV1 (%) postBD, as well as lower albeit not statistically significantly FEV1 (%) preBD (*p* = 0.051) (Table 2). Furthermore, T2-high asthma was characterized as a more severe, difficult-to-treat asthma, as it was uncontrolled despite the use of inhaled and oral corticosteroids, and T2-high asthma patients more often needed biological therapy to control their asthma symptoms/exacerbations. Since 15 of 24 (63%) of T2-high asthmatic needed biological treatment, either omalizumab or mepolizumab, while in the group of T2-low asthmatics only 2 of 12 (17%), had difficult-to-treat asthma, according to need for oral corticosteroids and only one was then eventually treated with biological therapy (Table 2).

Consequently, among the 16 patients that needed biological therapy to control their asthma symptoms/exacerbations, 15 (94%) had T2-high asthma, and only one (6%) had T2-low asthma (Figure 3). Our results demonstrate that patients with T2-high asthma had a more severe, difficult-to-treat asthma in comparison to T2-low asthma, in which better outcomes are achieved with conventional treatment.

## 3. Discussion

Herein we demonstrate that interleukins transcripts can be successfully detected in induced sputum from asthmatic patients. mRNA expression levels of *IL4*, *IL5,* and *IL13* are increased in sputum cells from asthmatic patients in comparison to controls and can be used as molecular biomarkers to categorize patients into T2-high and T2-low asthma endotypes similarly as described previously [5,11].

Using the previously described cut-off point for T2-high [5], we added 2 SDs to the mean T2 gene mean of control subjects and demonstrated that 67% of our asthmatic patients are T2-high. Similarly, in the previous report analyzing the same sputum gene signatures, it was reported that 70% of asthmatics is T2-high [5]. To our knowledge, this is the first report, which clinically adopted T2 sputum gene signatures and correlated this molecular profiling with the selection of adjunct biological therapy. Given that 94% of patients with the need for biological therapy demonstrated T2-high gene signature, sputum expression profiling should be considered clinically in all individuals with difficult-to-treat asthma. The clinical phenotype of the T2-high patients as compared to the T2-low patients was found to be more severe, with slightly reduced lung functions as determined by forced expiratory volume in 1 s (FEV1), and it was more often uncontrolled despite the use of inhaled and/or oral corticosteroids and T2-high asthma patients more often needed biological therapy to control their asthma symptoms/exacerbations. Findings that T2-high asthmatics had a more severe clinical phenotype are in agreement with previous studies using different molecular approaches for asthma classification, that reported that T2-high asthmatics had more severe disease in comparison to T2-low, determined as reduced lung functions [5,11,12], resistance to high-dose inhaled corticosteroids, requiring treatment with oral corticosteroids [10,11,12,13] and poorer asthma control assessed by different asthma control questionnaires [5,10,12,13,14]. Interestingly, among the patients with highly elevated sputum eosinophils (>6%), high FeNO (>32 ppb) and T2-high gene mean, all (6) patients have several exacerbations per year and the average FEV1 (%) preBD of those patients was 66.5%, in comparison to other patients in which only 57% (17/30) of patients experienced exacerbations and have average FEV1 (%) preBD of 85%. Similarly as demonstrated in our study, also others have found that T2-high asthma is a more severe, difficult-to-treat asthma often requiring biological treatment to control exacerbations/symptoms [13]. However, in contrast to our results where we have not found any differences between T2-high and T2-low in exhaled nitric oxide fraction (FeNO), others have reported that higher exhaled nitric oxide fraction is a typical characteristic of T2-high asthma [5,12]. Nevertheless, a recent study that analyzed different transcriptome-associated clusters, also reported that a typical T2-high cluster was not associated with exhaled nitric oxide fraction in comparison to other T2-low clusters [10]. Higher OCS maintenance use in T2-high asthma might lower FeNO and contribute to the discrepancies between different studies. Even though some studies reported that FeNO was found to be higher in T2-high asthma, our results are in line with the findings that FeNO is not a good per se predictor of T2-high asthma, and currently, there are no specific recommendations on how FeNO can help in selecting appropriate treatment options [1,12,15,16].

T2-high asthmatic patients had higher sputum eosinophil counts, which is the canonical marker of T2-high asthma [5,9,12,16], as well as blood eosinophil, which is in line with our findings. This indicates that eosinophilic and T2-mediated airway inflammation represents one of the most important traits in asthma, and in addition to Th2 cells, innate ILC-2 cells have been found in the airways of asthmatic patients [13,16,17,18]. Several cytokines have been implicated in the pathophysiology of T2-mediated, most notably IL-4, IL-5, and IL-13 [13]. The effects of those cytokines are the activation of airway epithelial cells, recruitment of different effector cells, including mast cells, basophils, and eosinophils, as well as epithelial and subepithelial remodeling, resulting in airway hyperresponsiveness [13,16]. T2 inflammation is evident in both allergic and nonallergic patients and this is in line with previous reports that approximately 50% of asthma patients have T2 inflammation without allergy [17].

Even though we have found increased T2 inflammation in two-thirds of asthmatics, 33 % of our asthmatics were classified as having T2-low asthma, which is a highly heterogeneous condition composed of multiple T2-low asthma phenotypes [10,14,17,19]. Several cells and mediators have been linked to T2-low asthma, including the activation of Th1 and Th17, neutrophil infiltration, and cytokines IL-8, IL-17, IL-21, and IL-22. However, no clinically applicable biomarkers and/or targeted treatment are currently available in T2-low asthma [9,13,17]. This high heterogeneity was also noted in our T2-low asthmatics. Nevertheless, only two of our 12 T2-low patients, had a more severe disease experiencing several exacerbations per year and low FEV1 (62% and 65%). Those observations supporting previous findings that only a minor subgroup of T2-low asthmatics had severe, difficult-to-treat asthma [10,16].

Several molecular approaches have been previously used to define T2-high asthma [5,10,12,17,19], using both unsupervised and supervised transcriptomic and proteomic approaches for asthma clustering or stratification. In this study, similarly as described by Peters et al. [5], we performed gene expression analysis in induced sputum cells focused on major T2-associated cytokines IL-4, IL-5, and IL-13. By implementing the T2 gene mean, a combined metric of gene expression of these three cytokines, we were able to stratify asthmatics into T2-high and T2-low asthma. This approach was demonstrated to be highly robust since the percentage and clinical characteristics of T2-high asthmatics are similar as described previously, using the same gene signature [5]. Measurement of gene expression of the most important cytokines that drive T2-inflammation, specifically IL-4, IL-5, and Il-13, in induced sputum could be applied as a reliable biomarker for the identification of T2-high asthma. In contrary to some previous reports describing that the expression levels of *IL4*, *IL5,* and *IL13* in sputum were extremely low determined by Affymetrix chip assay or by PCR [12,20], we have shown that using RT-qPCR mRNA expression of these cytokines can be robustly and reliably quantified. T2 gene mean was found to be stable over time in asthmatics and was not influenced by steroid treatment, what are the advantages of using sputum mRNA profiling over other methods to categorize patients into T2-high and T2-low asthma [5,11].

There are some limitations of our study. First, the small number of included subjects, especially in the control group. Additionally, a larger number of asthmatic patients would allow us to perform more subanalysis and we believe that some correlations would be even more evident. Another limitation of using gene expression profiling of selected cytokines *(IL4*, *IL5*, and *IL13*) in induced sputum for asthma stratification in comparison to unsupervised approaches employing several biomarkers is that prevalence of T2-high asthma might be underestimated. On the other hand, as described previously this approach is more robust, reproducible, and easier to implement into the clinic. Although we have shown that molecular profiling in induced sputum can be translated to clinical practice, the technique is complex and could only be performed in specialized and expert centers for specific cases in which other broadly used methods failed to classify patients as having T2-high or T2-low asthma. Currently, clinically applicable biomarkers, such as serum specific IgE, FeNO, and blood and sputum eosinophil counts, can to some extent help with the decision-making in selecting the most appropriate treatment options, including biologics [9,13,17]. However, these biomarkers are not specific enough to identify different asthma endotypes of individual patients, which is crucial to select the most suitable targeting treatment of the underlying inflammatory mechanism [9,13,17]. Therefore, the identification of novel and more specific biomarkers for asthma endotyping is the subject of intense research [13]. Our study demonstrated that gene expression profiling of *IL4*, *IL5,* and *IL13* in induced sputum can be used as molecular biomarkers to categorize patients into T2-high and T2-low endotypes. Moreover, T2-high asthma was characterized as more severe, difficult-to-treat asthma. Consequently, sputum gene expression profiling of T2-high/T2-low endotypes should improve the selection of targeted therapies and management of asthma patients.

## 4. Materials and Methods

### 4.1. Study Subjects

We prospectively recruited 36 patients with a diagnosis of asthma according to Global Initiative for Asthma (GINA) [1] treated at the University Clinic for Respiratory and Allergic Diseases Golnik in years 2015 and 2016. All included patients at the time of sampling underwent lung function tests (pre-and post-bronchodilator inhalation), methacholine challenge tests, measurement of nitric oxide in exhaled breath, skin prick tests, and/or serum IgE determination for common allergens, measurement of total serum IgE, and sputum induction. The reversibility test was defined as positive if FEV1 increased by 12% (200 mL) or more after the administration of 400 mg of salbutamol. All patients showed a positive methacholine test defined as a decrease of baseline FEV1 of 20% with a cumulative dose of methacholine (PD20) less than 4 mg. Asthma exacerbations were defined as exacerbation with the need for OCS use in the past year. Control subjects, which were sex and age-matched with asthmatics, had no known chronic diseases. The characteristics of the study groups are shown in Table 2. The study was conducted in accordance with the amended Declaration of Helsinki. It was approved by the Slovenian National Medical Ethics Committee (approval number 95/06/13), and all patients gave their informed written consent.

### 4.2. Sputum Collection and Processing

Sputum was induced and processed as described previously [5,21]. Briefly, subjects inhaled nebulized 4.5% hypertonic saline for three 5-min periods. To minimize saliva contamination, subjects were asked to mouthwash thoroughly and expectorate saliva into a separate container before producing sputum. The collected sputum was immediately processed. The volume of the entire sputum sample was determined, and an equal volume of 10% solution of Sputolysin (Calbiochem, San Diego, CA, USA) was added. The samples were then mixed and incubated for 15 min at 37 °C to ensure complete homogenization. The samples were then centrifuged at 4 °C 2000 rpm for 10 min and cell pellets were stored in QIAzol Lysis Reagent (Qiagen, Hilden, Germany) at –80 °C until RNA isolation. Cytospins were stained according to the May–Grünwald–Giemsa and Papanicolaou methods. Differential cell counts were performed by one observer, who counted 200 non-epithelial cells. Only samples with a quality score of 7 or higher were used for further analysis [21].

### 4.3. RNA Isolation and Gene Expression

Total RNA was isolated from sputum cells by using the miRNeasy Mini Kit (Qiagen) according to the manufacturer’s instructions. RNA quantity and quality were assessed using NanoDrop 2000c (Thermo Fisher Scientific, Waltham, MA, USA) and 2100 bioanalyzer (Agilent Technologies, Santa Clara, CA, USA). Following reverse transcription using a high-capacity cDNA Reverse Transcription Kit (Applied Biosystems, Foster City, CA, USA), cDNA was quantified by using real-time PCR (ABI PRISM 7500 Real-Time PCR System; Applied Biosystems) under standard conditions with TaqMan Fast Advanced Master Mix (Applied Biosystems). TaqMan assays IL4 (Hs00174122_m1), IL5 (Hs01548712_g1), and IL13 (Hs00174379_m1) were utilized to determine *IL4*, *IL5*, and *IL13* mRNA expression levels and glyceraldehyde-3-phosphate dehydrogenase (GAPDH) was used as an endogenous control (Applied Biosystems). A sample from the control group was used as a calibrator. All measurements were performed in triplicate for each sample, and relative expression was analyzed using the ΔΔCt method. “T2 gene mean” was calculated as the mean of standardized gene expression variables of *IL4*, *IL5*, and *IL13*, as described previously [5].

### 4.4. Statistical Analyses

The distribution of data was determined using the D’Agostino and Pearson omnibus normality test. The strength of association between different parameters was analyzed with the Mann–Whitney U-test, unpaired *t*-test, Fisher’s exact test, or one-way ANOVA test followed by Holm–Sidak multiple comparison test as appropriate. The Spearman rank correlation test was used to analyze the degree of linear association between analyzed parameters. The cut-off value between low and high T2 gene mean (to classify patients as T2-high and T2-low), was calculated by adding two SDs to the mean T2 gene mean of control subjects, as described previously [5]. Statistical analyses were performed using GraphPad Prism 8 software (San Diego, CA, USA), and probability values of *p* < 0.05 were accepted as significant.

## 5. Conclusions

In conclusion, we demonstrate that interleukins transcripts can be robustly and efficiently detected in induced sputum from asthmatic patients. mRNA expression levels of *IL4*, *IL5,* and *IL13* are increased in sputum cells from asthmatic patients in comparison to controls. T2 gene mean, a combined metric of gene expression of these three cytokines, can be used as molecular biomarkers to categorize patients into T2-high and T2-low asthma. T2-high endotype is characterized by eosinophilia, and severe, difficult-to-treat asthma, which often requiring biological treatment to control exacerbations/symptoms.

## Figures and Tables

**Figure 1 life-11-00092-f001:**
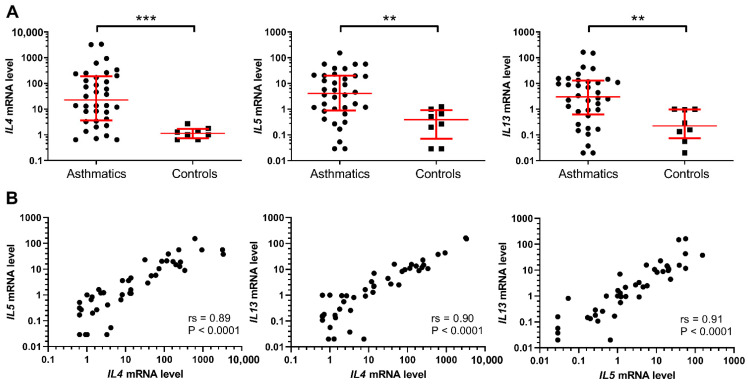
(**A**) Gene expression levels of *IL4*, *IL5*, and *IL13* were significantly increased in asthmatic patients’ samples compared to controls. Mann–Whitney U-test: ** *p* < 0.01; *** *p* < 0.001. (**B**) Spearman’s rank correlation analysis of associations between *IL4*, *IL5*, and *IL13* mRNA expression levels.

**Figure 2 life-11-00092-f002:**
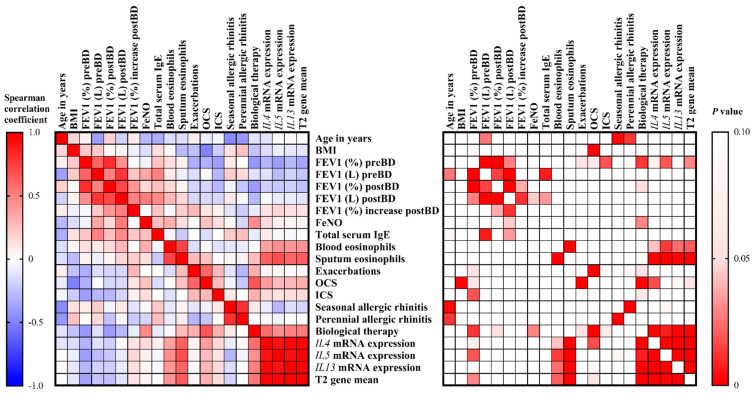
Correlation heatmap reporting Spearman correlation coefficients and P values for each comparison. The bar on the left side of the map indicates the color legend of the Spearman correlation coefficients, and the bar on the right side of the map indicates the color legend of the P values calculated for each pair of samples in the matrix. The Spearman rank correlation test was used. A *p* value < 0.05 was accepted as significant. *p* > 0.10 are presented in white squares.

**Figure 3 life-11-00092-f003:**
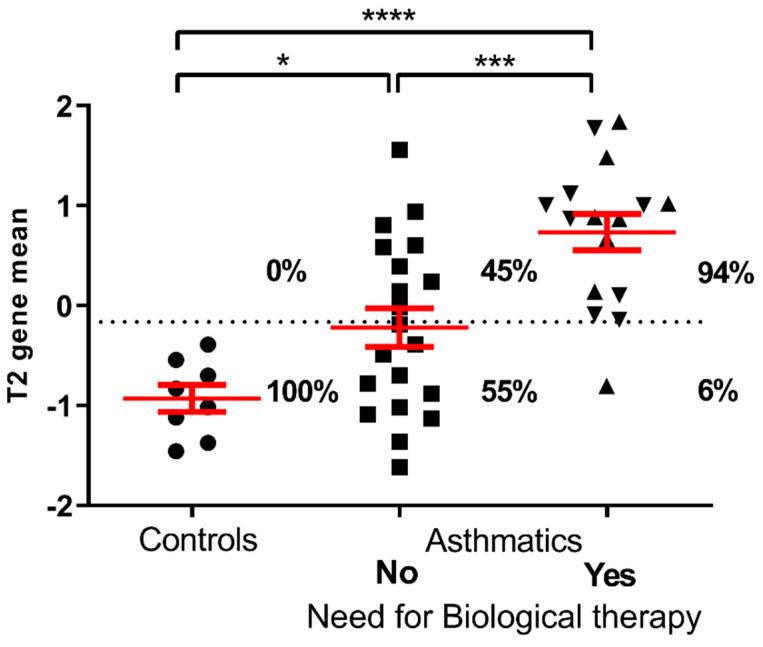
T2 gene mean calculated by combining the expression levels of *IL4*, *IL5,* and *IL13* in controls and asthmatics divided by need for biological therapy either omalizumab (▼) or mepolizumab (▲). The dotted line represents the T2 gene mean cut-off point for classification into T2-high/T2-low asthma. ANOVA test followed by Holm-Sidak multiple comparison test: * *p* < 0.05; *** *p* < 0.001; **** *p* < 0.0001.

**Table 1 life-11-00092-t001:** Differences in clinical and laboratory characteristics between T2-high and T2-low asthmatic patients.

	T2-low Asthma	T2-High Asthma	*p* Value ^b^
No.	12	24	
Age in years, median (IQR)	59 (27.0)	53 (19.3)	0.779
Female sex, No. (%)	9 (75)	17 (71)	1.000
BMI (kg/m^2^), median (IQR)	25.2 (4.9)	27.5 (5.7)	0.748
FEV_1_ (%) preBD, median (IQR)	92.5 (39.3)	70 (28.3)	0.051
FEV_1_ (L) preBD, median (IQR)	2.43 (0.92)	2.20 (1.02)	0.366
FEV_1_ (%) postBD, median (IQR)	97 (38)	77 (25)	**0.019**
FEV_1_ (L) postBD, median (IQR)	2.99 (1.41)	2.21 (1.31)	0.050
FEV1 (%) increase postBD, median (IQR)	9 (16)	7 (15.5)	0.391
Positive reversibility test, No. (%)	2 (17)	4 (17)	1.000
FeNO (ppb), median (IQR)	44 (57)	55 (69)	0.365
Total serum IgE (kU/L), median (IQR)	338 (871)	136 (261)	0.449
Blood eosinophils (cells/μL), median (IQR)	205 (253)	535 (855)	**0.035**
Sputum eosinophils (%), median (IQR)	5 (11.5)	18 (42.0)	**0.017**
Active/ex-smokers, No. (%)	1/3 (8.3/25.0)	2/4 (8.3/16.7)	0.700
Exacerbations in the past year, No. (%)	8 (67)	15 (63)	1.000
ICS use, No. (%)	10 (83)	22 (92)	0.588
OCS maintenance use, No. (%)	2 (17)	12 (50)	0.076
Seasonal allergic rhinitis, No. (%)	6 (50)	8 (33)	0.472
Perennial allergic rhinitis, No. (%)	6 (50)	12 (50)	1.000
Biological therapy ^a^, No. (%)	1 (8)	15 (63)	**0.004**

Abbreviations: BMI, body mass index; FEV1, forced expiratory volume in 1 s; FeNO, exhaled nitric oxide; ICS, inhaled corticosteroids; preBD, pre-bronchodilation; postBD, post-bronchodilation; OCS, oral corticosteroids. ^a^ Omalizumab (eight patients) or Mepolizumab (eight patients). ^b^ Statistically significant *p* values are in boldface.

**Table 2 life-11-00092-t002:** Demographic, clinical, and laboratory characteristics of control subjects and asthmatic patients.

	Control Subjects	Asthmatic Subjects	*p* Value ^b^
No.	8	36	
Age in years, median (IQR)	48.5 (14.3)	55.5 (19.5)	0.223
Female sex, No. (%)	7 (88)	26 (72)	0.656
BMI (kg/m^2^), median (IQR)	26.9 (7.9)	26.1 (6.2)	0.899
FEV_1_ (%) preBD, median (IQR)	100.5 (11.8)	74 (42.1)	**0.017**
FEV_1_ (L) preBD, median (IQR)	2.88 (0.69)	2.33 (1.01)	**0.013**
FEV_1_ (%) postBD, median (IQR)	ND	2.26 (1.31)	
FEV_1_ (L) postBD, median (IQR)	ND	79 (29)	
Positive reversibility test, No. (%)	ND	6 (17)	
FeNO (ppb), median (IQR)	30 (54)	55 (57)	0.313
Total serum IgE (kU/l), median (IQR)	ND	168 (417)	
Blood eosinophils (cells/μL), median (IQR)	180 (65)	315 (720)	0.197
Sputum eosinophils (%), median (IQR)	0.5 (4.8)	16 (25.5)	**0.016**
Active/ex-smokers, No. (%)	4/0 (50/0)	3/7 (8.3/19.4)	0.242
Exacerbations in the past year, No. (%)		23 (64)	
ICS ^c^ use, No. (%)		32 (89)^c^	
OCS maintenance use, No. (%)		14 (39)	
Seasonal allergic rhinitis, No. (%)		14 (39)	
Perennial allergic rhinitis, No. (%)		18 (50)	
Biological therapy ^a^, No. (%)		16 (44)	

Abbreviations: BMI, body mass index; FEV1, forced expiratory volume in 1 s; FeNO, exhaled nitric oxide; ICS, inhaled corticosteroids; preBD, pre-bronchodilation; postBD, post-bronchodilation; OCS, oral corticosteroids. ^a^ Omalizumab (eight patients) or Mepolizumab (eight patients). ^b^ Statistically significant *p* values are in boldface. ^c^ ICS dose was equivalent to the median (IQR) 1000 (1010) mg of budesonide daily. Twenty-nine (81%) patients were taking ICS/LABA.

## Data Availability

Data are contained within the article.

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
