# Peer review of "T2-high Asthma, Classified by Sputum mRNA Expression of IL4, IL5, and IL13, is Characterized by Eosinophilia and Severe Phenotype"

_life, 2021, doi:10.3390/life11020092_

Round 1

Reviewer 1 Report

General Comments:

Rijavec et al provide an interesting and potentially helpful addition to the literature with this study which assessed the feasibility of using sputum m-RNA expression of relevant cytokines to classify patients with T2-high asthma. The Introduction is nicely compiled as is the Discussion, while the Methods section and Results section are clearly presented. The methods employed in the study are generally appropriate. The Golnik clinic is widely acknowledged for its expertise in induced sputum work and that is well-reflected in this study. The definition of T2 high by T2 gene mean cut-off is very reasonable and easier to replicate than many of the current loose definitions of T2 status available to clinicians. The findings are plausible and intuitive, demonstrating significantly elevated T2-cytokine expression in asthmatic subjects, with good correlation in expression of the studied cytokines. Using their bespoke cut-off for T2 high status based on mean gene expression they go on to demonstrate that 2/3 of the asthmatic subjects studied were T2 high. In support of their classification, these T2 high subjects also had higher eosinophil expression in blood and sputum. Of note, the T2 high subjects showed some basic signals for worse asthma severity which is consistent with a growing consensus of opinion that the majority of subjects with difficult-to-treat asthma have a T2 phenotype. They conclude that this approach using T2 gene mean could be used as a sputum biomarker for T2 high disease in patients with asthma.

This work merits publication. However, I have some comments about necessary revisions which should enhance it, prior to acceptance..

Specific Comments:

  1. The sections of the paper are nicely written but I find the current order of the sections unhelpful and somewhat confusing to the reader. Can they please present the sections in conventional order – Introduction, Materials/ Methods, Results, Discussion, Conclusion.
  2. The authors have studied 36 asthma patients but only 8 controls. In Table 1, they present comparative characterisation of these 2 groups which demonstrate their broad similarities. That is reassuring but the discrepancy in sample size between these groups is a potential limitation of the study. That should be mentioned in the Discussion.
  3. In line with point (2), the Discussion would benefit from a paragraph summarising strengths and limitations of the work to aid interpretation of the relevance of findings.
  4. In the Methods, can they please clarify whether spirometry, blood eosinophil measurement, sputum eosinophil measurement and FENO measurement were recorded at the same timepoint or not? This is relevant since there is an inherent variability to these measures.
  5. In section 2.2 and Table 2, the Authors present a comparison of the T2 v non-T2 asthmatics based on their classification of T2 status. This is interesting for practicing clinicians. Some clarity on the terms used with definitions is needed to aid understanding. Specifically;
  6. Is FEV1 pre-bronchodilator or post-bronchodilator or just done in clinical practice (what might be called “clinic FEV1”)?
  7. What is the timeframe for exacerbations? For instance, “in the last year”? Also, are these exacerbations needing OCS or can it be milder?
  8. What is meant by OCS use? Is this need for courses of OCS or need for daily maintenance OCS, or either? These levels of OCS use reflect different dimensions of poor asthma control. If possible it would be best to separate into the different forms of OCS use.
  9. What is meant by “allergies” in seasonal and year-round allergies? Please explain in the table footnote – is it just rhinitis? If so better to state rhinitis. If it encompasses other states such as eczema, food allergy etc then explain what is covered by the term.
  10. Some more detail on indices of severity would be helpful, if the Authors have it. See points 11-15.
  11. Is there data on ICS dose as well?
  12. Is there data on GINA step therapy?
  13. Is there data on hospital admissions?
  14. Is there data on needing ICU admission?
  15. Is there data such as ACQ or ACT?
  16. Was the prevalence of current smoking similar for the T2/ non-T2 group? Smoking of course might have influenced both FEV1 and FeNO findings in Table 2.
  17. In the Discussion, the Authors discuss that FENO did not differ significantly between T2 and non-2 asthma (lines 140-149). A point that they may wish to consider in the Discussion is whether this reflects “confounding by indication”. For example, if T2 subjects have genuinely more severe asthma and have higher ICS dosage needs, maintenance OCS use and/or Biologics use then the similar FeNO in T2 subjects (compared to non-T2) may partly reflect the FeNO lowering effects of those treatments?
  18. The high prevalence of T2 status in this study mirrors that in the previous Peters JACI paper using similar classification and is in line with other emerging clinical classifications of T2 status. Indeed compared to some of those clinical classifications, T2 status in the present Golnik clinic study may even be lower than expected. Could that reflect a limitation of a one dimensional perspective on T2 status, solely reliant on sputum m-RNA expression on selected cytokines? Might be worth reflecting on in the Discussion.
  19. There is significant temporal longitudinal variability to expression of features of T2 inflammation such as blood or sputum eosinophils as well as FeNO. That makes single timepoint cross-sectional assessment of T2 status in asthma less reliable than a broader longitudinal perspective. Do the Authors feel that their one timepoint m-RNA expression of T2 cytokines is an improvement on those one timepoint measures. Again would be worth discussing.
  20. The Authors have shown that this technique of T2 classification using sputum is feasible and seem to suggest that this could be readily used in clinical practice. The reality is though, that unlike the Golnik clinic, many Severe Asthma clinics struggle to perform sputum induction within their standard clinical practice. That point should be noted in the Discussion. One attraction of this technique would be in cases of diagnostic doubt where, in more specialist centres, it could be used to seek evidence for T2 status where other clinically detectable T2 signals (blood eosinophils, FeNO etc) are not seen (perhaps due to levels of treatment such as daily OCS). Again might be worth raising in the Discussion.

Author Response

Response to Reviewer 1 Comments

General Comments:

Rijavec et al provide an interesting and potentially helpful addition to the literature with this study which assessed the feasibility of using sputum m-RNA expression of relevant cytokines to classify patients with T2-high asthma. The Introduction is nicely compiled as is the Discussion, while the Methods section and Results section are clearly presented. The methods employed in the study are generally appropriate. The Golnik clinic is widely acknowledged for its expertise in induced sputum work and that is well-reflected in this study. The definition of T2 high by T2 gene mean cut-off is very reasonable and easier to replicate than many of the current loose definitions of T2 status available to clinicians. The findings are plausible and intuitive, demonstrating significantly elevated T2-cytokine expression in asthmatic subjects, with good correlation in expression of the studied cytokines. Using their bespoke cut-off for T2 high status based on mean gene expression they go on to demonstrate that 2/3 of the asthmatic subjects studied were T2 high. In support of their classification, these T2 high subjects also had higher eosinophil expression in blood and sputum. Of note, the T2 high subjects showed some basic signals for worse asthma severity which is consistent with a growing consensus of opinion that the majority of subjects with difficult-to-treat asthma have a T2 phenotype. They conclude that this approach using T2 gene mean could be used as a sputum biomarker for T2 high disease in patients with asthma.

This work merits publication. However, I have some comments about necessary revisions which should enhance it, prior to acceptance..

Specific Comments:

Point 1: The sections of the paper are nicely written but I find the current order of the sections unhelpful and somewhat confusing to the reader. Can they please present the sections in conventional order – Introduction, Materials/ Methods, Results, Discussion, Conclusion.

Response 1: The order of the sections is made according to the Journal instructions and therefore should be kept as it is.

Point 2: The authors have studied 36 asthma patients but only 8 controls. In Table 1, they present comparative characterisation of these 2 groups which demonstrate their broad similarities. That is reassuring but the discrepancy in sample size between these groups is a potential limitation of the study. That should be mentioned in the Discussion.

Point 3: In line with point (2), the Discussion would benefit from a paragraph summarising strengths and limitations of the work to aid interpretation of the relevance of findings.

Response 2 & 3: Thank you for your careful examination of our manuscript. We have added in the Discission section (lines 201-212) a paragraph explaining the limitation concerning sample size and other limitations and strengths of our study »There are some limitations of our study. First, the small number of included subjects, especially in the control group. Besides, a larger number of asthmatic patients would allow us to perform more subanalysis and we believe that some correlations would be even more evident. Another limitation of using gene expression profiling of selected cytokines (IL4, IL5, and IL13) in induced sputum for asthma stratification in comparison to unsupervised approaches employing several biomarkers is that prevalence of T2-high asthma might be underestimated. On the other hand, as described previously this approach is more robust, reproducible, and easier to implement into the clinic. Although we have shown that molecular profiling in induced sputum can be translated to clinical practice, the technique is complex and could only be performed in specialized and expert centers for specific cases in which other broadly used methods failed to classify patients as having T2-high or T2-low asthma.«

Furthermore, we have also inserted another paragraph in the Discussion section (lines 127-132) highlighting the most important findigs of our study »To our knowledge, this is the first report, which clinically adopted T2 sputum gene signatures and correlated this molecular profiling with the selection of adjunct biological therapy. Given that 94% of patients with the need for biological therapy demonstrated T2-high gene signature, sputum expression profiling should be considered clinically in all individuals with difficult-to-treat asthma.«

Point 4: In the Methods, can they please clarify whether spirometry, blood eosinophil measurement, sputum eosinophil measurement and FENO measurement were recorded at the same timepoint or not? This is relevant since there is an inherent variability to these measures.

Response 3: Thank you for this comment. All measurements were determined at the same time point as it is stated in the Methods section (lines 230-234) »All included patients at the time of sampling underwent lung function tests (pre-and post-bronchodilator inhalation), methacholine challenge tests, measurement of nitric oxide in exhaled breath, skin prick tests, and/or serum IgE determination for common allergens, measurement of total serum IgE, and sputum induction.«

Point 5: In section 2.2 and Table 2, the Authors present a comparison of the T2 v non-T2 asthmatics based on their classification of T2 status. This is interesting for practicing clinicians. Some clarity on the terms used with definitions is needed to aid understanding. Specifically;

Point 6: Is FEV1 pre-bronchodilator or post-bronchodilator or just done in clinical practice (what might be called “clinic FEV1”)?

Response 5-6: FEV1 in the original manuscript was pre-bronchodilatation. In the revised manuscript we have also added FEV1 post-bronchodilatation. For both measurements, we have absolute values in L, as well as percentages (%).

We have corrected in Table 1, Table 2, and in abbreviations list stating that FEV1 (%, L) preBD is defined as pre-bronchodilation forced expiratory volume in 1 s and FEV1 (%, L) postBD is defined as post-bronchodilation forced expiratory volume in 1 s.

Point 7: What is the timeframe for exacerbations? For instance, “in the last year”? Also, are these exacerbations needing OCS or can it be milder?

Response 7: The timeframe for exacerbations was in the last year from the sampling. We have inserted a sentence explaining this in the Methods section (lines 237-238) »Asthma exacerbations were defined as exacerbation with the need for OCS use in the past year.« We have also corrected Table 1 and Table 2 indicating the timeframe for exacerbations, stating »Exacerbations in the past year«.

Point 8: What is meant by OCS use? Is this need for courses of OCS or need for daily maintenance OCS, or either? These levels of OCS use reflect different dimensions of poor asthma control. If possible it would be best to separate into the different forms of OCS use.

Response 8: Thank you for this comment. »OCS use« means a need for maintenance daily use and not courses of OCS. We have included this comment in the table and change the terminology to »OCS maintenance use«.

Point 9: What is meant by “allergies” in seasonal and year-round allergies? Please explain in the table footnote – is it just rhinitis? If so better to state rhinitis. If it encompasses other states such as eczema, food allergy etc then explain what is covered by the term.

Response 9: The term »allergies« means seasonal and year-round allergies and clinically rhinitis. We have changed the phrasing to »seasonal allergic rhinitis« and »perennial allergic rhinitis« in Table 1, Table 2, and Figure 2.

Point 10: Some more detail on indices of severity would be helpful, if the Authors have it. See points 11-15.

Point 11: Is there data on ICS dose as well?

Point 12: Is there data on GINA step therapy?

Point 13: Is there data on hospital admissions?

Point 14: Is there data on needing ICU admission?

Point 15: Is there data such as ACQ or ACT?

Response 10-15: Unfortunately we were not able to obtain that information as these data were not available from the digital medical documentation.

Point 16: Was the prevalence of current smoking similar for the T2/ non-T2 group? Smoking of course might have influenced both FEV1 and FeNO findings in Table 2.

Response 16: Thank you for this suggestion. We have added the smoking status to Table 1 and Table 2. The prevalence of active and ex-smokers was similar in T2-low and T2-high groups (Table 2).

Point 17: In the Discussion, the Authors discuss that FENO did not differ significantly between T2 and non-2 asthma (lines 140-149). A point that they may wish to consider in the Discussion is whether this reflects “confounding by indication”. For example, if T2 subjects have genuinely more severe asthma and have higher ICS dosage needs, maintenance OCS use and/or Biologics use then the similar FeNO in T2 subjects (compared to non-T2) may partly reflect the FeNO lowering effects of those treatments?

Response 17: None of the patients received biological therapy at the time of sampling. However, patients have indeed received ICS/OCS before sampling and this might affect FeNO.

We have added this limitation and explanation in the Discussion section (lines 154-156) explaining that »Higher OCS maintenance use in T2-high asthma might lower FeNO and contribute to the discrepancies between different studies.«

Point 18: The high prevalence of T2 status in this study mirrors that in the previous Peters JACI paper using similar classification and is in line with other emerging clinical classifications of T2 status. Indeed compared to some of those clinical classifications, T2 status in the present Golnik clinic study may even be lower than expected. Could that reflect a limitation of a one dimensional perspective on T2 status, solely reliant on sputum m-RNA expression on selected cytokines? Might be worth reflecting on in the Discussion.

Response 18: We have added this in the Discussion section also this limitation (lines 204-208) »Another limitation of using gene expression profiling of selected cytokines (IL4, IL5, and IL13) in induced sputum for asthma stratification in comparison to unsupervised approaches employing several biomarkers is that prevalence of T2-high asthma might be underestimated. On the other hand, as described previously this approach is more robust, reproducible, and easier to implement into the clinic.«

Point 19: There is significant temporal longitudinal variability to expression of features of T2 inflammation such as blood or sputum eosinophils as well as FeNO. That makes single timepoint cross-sectional assessment of T2 status in asthma less reliable than a broader longitudinal perspective. Do the Authors feel that their one timepoint m-RNA expression of T2 cytokines is an improvement on those one timepoint measures. Again would be worth discussing.

Response 19: Thank you for raising this important issue. Peters et al., 2013 reported that T2 gene mean values tended to be stable over time in asthmatic patients.

We have added this in the Discussion section (lines 197-200) »T2 gene mean was found to be stable over time in asthmatics and was not influenced by steroid treatment, what are the advantages of using sputum mRNA profiling over other methods to categorize patients into T2-high and T2-low asthma [5, 11].«

Point 20: The Authors have shown that this technique of T2 classification using sputum is feasible and seem to suggest that this could be readily used in clinical practice. The reality is though, that unlike the Golnik clinic, many Severe Asthma clinics struggle to perform sputum induction within their standard clinical practice. That point should be noted in the Discussion. One attraction of this technique would be in cases of diagnostic doubt where, in more specialist centres, it could be used to seek evidence for T2 status where other clinically detectable T2 signals (blood eosinophils, FeNO etc) are not seen (perhaps due to levels of treatment such as daily OCS). Again might be worth raising in the Discussion.

Response 20: We have added this in the Discussion section (lines 208-212) »Although we have shown that molecular profiling in induced sputum can be translated to clinical practice, the technique is complex and could only be performed in specialized and expert centers for specific cases in which other broadly used methods failed to classify patients as having T2-high or T2-low asthma.«

Reviewer 2 Report

The present manuscript " T2-high asthma, classified by sputum mRNA expression of IL4,IL5, and IL13, is characterized by eosinophilia and severe phenotype" used the mRNA expression of T2 cytokines IL4, IL5, and IL13 in sputum as a biomarker for T2 inflammation and correlate them with several clinical characteristics.  In particular, the authors have shown a positive correlation between the presence of a T2 signature (expressed as high mRNA expression of IL4, IL5 and IL13), with an eosinophilic inflammation and a high severity.  The classification of asthma patients in T2high and T2low phenotypes is currently the most popular classification of asthmatics between clinicians. Indeed, it is based on that classification how clinicians select patients to undergo biological therapies and other asthma treatment strategies. However, due to the high complexity of asthma pathogenesis and pathophysiology, including different subpopulations of innate and adaptive cells and mediators, further studies are needed in order to specifically select subgroups of patients with a potentially high response to the different new biologic therapies. The study design, as well as the results are clear and well written. However, I have to reject the present manuscript in the present form for the following reason:

  1. The majority of the results the authors are showing in this work are not new and were already published by Peters MC and colaborators in 2014 (J. Allergy Clin Immunol. 2014 Feb;133(2):388-94. doi: 10.1016/j.jaci.2013.07.036.). In this work, Peters et al. have already used the expression of IL4, IL5 and IL13 to classify patients in T2high and T2low asthmatics. They have also demonstrated that those patients with T2high asthma had an increased blood and sputum eosinophilia and they were patients with a higher severity degree. They also described a lower asthma control (ACT levels) in patients with T2high asthma.
  2. The only new in the present manuscript is, therefore, figure 3, were authors evidence that patients with T2high asthma correlate with the ones using biological therapy. However, in biological therapy authors only include Omalizumab (anti-IgE) and Mepolizumab (anti-IL5). Therefore, the article makes no original significant contribution. I kindly encourage the authors to include patients with different biological therapies (others than only omalizumab and mepolizumab, such as dupilumab, lebrikizumab, benralizumab, reslizumab or others). As well, the authors should change the title and focus on the need for biological therapy and the correlation with T2 signature, as this is their contribution.

Other suggestions:

  1. Include Asthma control questionnaire, as well as methacholine PD20 in the table. How asthma control and AHR differ in T2high and T2low asthma.
  2. Include allergic status of the patients, subclassification based on T2high allergic/non-allergic and T2low allergic/non-allergic.
  3. How do you performed AACt method? Do you use the mean value of healthy controls as reference? Please explain.
  4. Please indicate the statistical test used in each figure
  5. Figure 1. Remove the line in the correlation graph. When you use Spearman (non-parametric) you cannot represent a linear correlation. Keep only the dots.
  6. Please subclassified patients based on the biological therapy (i.e. omalizumab and mepolizumab) in order to assess the relation to T2 signature. It is mepolizumab only correlated with IL5 mRNA expression or also with IL4 and IL13? You can include this in the supplementary information.

Author Response

Response to Reviewer 2 Comments

The present manuscript " T2-high asthma, classified by sputum mRNA expression of IL4,IL5, and IL13, is characterized by eosinophilia and severe phenotype" used the mRNA expression of T2 cytokines IL4, IL5, and IL13 in sputum as a biomarker for T2 inflammation and correlate them with several clinical characteristics.  In particular, the authors have shown a positive correlation between the presence of a T2 signature (expressed as high mRNA expression of IL4, IL5 and IL13), with an eosinophilic inflammation and a high severity.  The classification of asthma patients in T2high and T2low phenotypes is currently the most popular classification of asthmatics between clinicians. Indeed, it is based on that classification how clinicians select patients to undergo biological therapies and other asthma treatment strategies. However, due to the high complexity of asthma pathogenesis and pathophysiology, including different subpopulations of innate and adaptive cells and mediators, further studies are needed in order to specifically select subgroups of patients with a potentially high response to the different new biologic therapies. The study design, as well as the results are clear and well written. However, I have to reject the present manuscript in the present form for the following reason:

Point 1: The majority of the results the authors are showing in this work are not new and were already published by Peters MC and colaborators in 2014 (J. Allergy Clin Immunol. 2014 Feb;133(2):388-94. doi: 10.1016/j.jaci.2013.07.036.). In this work, Peters et al. have already used the expression of IL4, IL5 and IL13 to classify patients in T2high and T2low asthmatics. They have also demonstrated that those patients with T2high asthma had an increased blood and sputum eosinophilia and they were patients with a higher severity degree. They also described a lower asthma control (ACT levels) in patients with T2high asthma.

Response 1: Thank you for your careful examination of our manuscript. We have added a paragraph in the Discussion section (lines 127-132) further highlighting what are the major novelties of our study »To our knowledge, this is the first report, which clinically adopted T2 sputum gene signatures and correlated this molecular profiling with the selection of adjunct biological therapy. Given that 94% of patients with the need for biological therapy demonstrated T2-high gene signature, sputum expression profiling should be considered clinically in all individuals with difficult-to-treat asthma.«

Point 2: The only new in the present manuscript is, therefore, figure 3, were authors evidence that patients with T2high asthma correlate with the ones using biological therapy. However, in biological therapy authors only include Omalizumab (anti-IgE) and Mepolizumab (anti-IL5). Therefore, the article makes no original significant contribution. I kindly encourage the authors to include patients with different biological therapies (others than only omalizumab and mepolizumab, such as dupilumab, lebrikizumab, benralizumab, reslizumab or others). As well, the authors should change the title and focus on the need for biological therapy and the correlation with T2 signature, as this is their contribution.

Response 2: Thank you for this comment. At the time of the study (2015, 2016) no other biological therapy for asthma was available in Slovenia, therefore we focused only on Omalizumab and Mepolizumab.

Other suggestions:

Point 3: Include Asthma control questionnaire, as well as methacholine PD20 in the table. How asthma control and AHR differ in T2high and T2low asthma.

Response 3: Unfortunately we were not able to obtain this information as ACT data are not available from the digital medical documentation of patients. Methacholine PD20 was performed earlier in the course of asthma diagnostic.

We have therefore upgraded the data of functional diagnostic with pre-and post-bronchodilator inhalation pulmonary function test (with data on FEV1 absolutely and in %).

Point 4: Include allergic status of the patients, subclassification based on T2high allergic/non-allergic and T2low allergic/non-allergic.

Response 4: If we stratify asthmatic patients in our study into four groups (T2-high allergic/non-allergic and T2-low allergic/non-allergic) the number of included patients in T2-low allergic/non-allergic groups are rather small, each group containing 6 patients, and therefore the results are not very informative. No differences were found between T2-high allergic and T2-high non-allergic as well as between T2-low allergic and T2-low non-allergic groups (data not shown).

Point 5: How do you performed AACt method? Do you use the mean value of healthy controls as reference? Please explain.

Response 5: We have added a sentence in the Methods section / 4.2. RNA isolation and gene expression (lines 271-272) explaining that we used a sample from the control group as a reference. »A sample from the control group was used as a calibrator.«

Point 6: Please indicate the statistical test used in each figure.

Response 6: In the legend of all figures we have inserted the explanation of statistical tests used.

Point 7: Figure 1. Remove the line in the correlation graph. When you use Spearman (non-parametric) you cannot represent a linear correlation. Keep only the dots.

Response 7: We have removed the line representing the linear correlation from Figure 1.

Point 8: Please subclassified patients based on the biological therapy (i.e. omalizumab and mepolizumab) in order to assess the relation to T2 signature. It is mepolizumab only correlated with IL5 mRNA expression or also with IL4 and IL13? You can include this in the supplementary information.

Response 8: Thank you for this comment. We have added (Table 1 & Table) 2 the number of patients that needed different biological treatment, specifically Omalizumab (8 patients) and Mepolizumab (8 patients). We have also marked the patients in Figure 3 that needed biological therapy as recipients of omalizumab (â–¼) or mepolizumab (â–²). There are no statistically significant differences in the T2 gene mean between recipients of omalizumab or mepolizumab. Unfortunately, the number of patients on mepolizumab is rather small (8 patients) to allow us any valid correlation analysis as we don't have much power to detect statistically significant correlations.

Reviewer 3 Report

In this study, the authors aimed to determine whether induced sputum samples can be used for gene expression profiling of T2-high asthma classified by IL4, IL5, and IL13 expression. Gene expression levels of IL4, IL5, and IL13 were significantly increased in asthmatic patients’ samples compared to controls. This manuscript is well written and the result is quite clear. However, I have several comments.

Major comments

  1. Important information regarding asthma is missing in this study. Please add the result of bronchodilator response test. Also, please add the percentage of patients with positive for bronchodilator test. Moreover, smoking state is also missing. Please add.

  1. Median value of FeNO in control was 30 ppb. This is too high. I am wondering if the control patients are normal subject.

  1. Most of the results of this study are already known. What is the novel finding of this study compared with previous study?

  1. The definition of exacerbation is not described in this manuscript. How did the authors identify the exacerbation? Prospectively or retrospectively? What is the period? Was it counted during one year?

  1. ICS was not used in 11% of patients. What’s the reason? Why ICS was not prescribed in some asthma patients?

Minor comments

  1. Please add FEV1 (L), FVC (L), FVC (%), and FEV1/FVC (%).

  1. Please specify whether lung function is pre or postbronchodilator value.

  1. Control state and ACT score are missing. Please add.

  1. How many patients are on ICSLABA? Please add.

  1. How many patients are on ICSLABALAMA? Please add.

Author Response

Response to Reviewer 3 Comments

In this study, the authors aimed to determine whether induced sputum samples can be used for gene expression profiling of T2-high asthma classified by IL4, IL5, and IL13 expression. Gene expression levels of IL4, IL5, and IL13 were significantly increased in asthmatic patients’ samples compared to controls. This manuscript is well written and the result is quite clear. However, I have several comments.

Major comments

Point 1: Important information regarding asthma is missing in this study. Please add the result of bronchodilator response test. Also, please add the percentage of patients with positive for bronchodilator test. Moreover, smoking state is also missing. Please add.

Response 1: Thank you for this comment. We have upgraded the data of functional diagnostic with pre-and post-bronchodilator inhalation pulmonary function test (with data on FEV1 absolutely and in %). In the Methods section (lines 234-235), we have added the description stating that »The reversibility test was defined as positive if FEV1 increased by 12% (200 mL) or more after the administration of 400 mg of salbutamol.« Furthermore, we also added the information on the percentage of patients with positive bronchodilator test (positive reversibility test) in Table 1 and Table 2.

We have also added the smoking status to Table 1 and Table 2.

The prevalence of patients with positive reversibility tests as well as the prevalence of active and ex-smokers was similar between T2-low and T2-high groups (Table 2).

Point 2: Median value of FeNO in control was 30 ppb. This is too high. I am wondering if the control patients are normal subject.

Response 2: Control subjects had no known chronic diseases, no respiratory symptoms and therefore no asthma symptoms.

According to NO measurement recommendation the interpretation of FeNO 30 ppb level includes:

»Intermediate FeNO (between 25 ppb and 50 ppb in adults; 20–35 ppb in children). The above data indicate that for FeNO values between 25 and 50 ppb, cautious interpretation is required. The weight placed on a FeNO result within this range will depend on whether the test is being used diagnostically in symptomatic steroid-naive subject, or whether the patient’s FeNO has increased or decreased from a previous value by what is deemed to be a clinically significant amount in a patient who is being monitored over time«.

At any one time, however, the most important consideration is whether or not the patient has current respiratory symptoms or a prior diagnosis of airways disease«

Reference:

Raed A. Dweik, Peter B. Boggs, Serpil C. Erzurum et al. An Official ATS Clinical Practice Guideline: Interpretation of Exhaled Nitric Oxide Levels (FENO) for Clinical Applications. Am J Respir Crit Care Med Vol 184. pp 602–615, 2011.

Point 3: Most of the results of this study are already known. What is the novel finding of this study compared with previous study?

Response 3: We have added a paragraph in the Discussion section (lines 127-132) further highlighting what are the major novelties of our study »To our knowledge, this is the first report, which clinically adopted T2 sputum gene signatures and correlated this molecular profiling with the selection of adjunct biological therapy. Given that 94% of patients with the need for biological therapy demonstrated T2-high gene signature, sputum expression profiling should be considered clinically in all individuals with difficult-to-treat asthma.«

Point 4: The definition of exacerbation is not described in this manuscript. How did the authors identify the exacerbation? Prospectively or retrospectively? What is the period? Was it counted during one year?

Response 4: The timeframe for exacerbations was in the last year from the sampling. We have inserted a sentence explaining this in the Methods section (lines 237-238) »Asthma exacerbations were defined as exacerbation with the need for OCS use in the past year« We have also corrected Table 1 and Table 2 indicating the timeframe for exacerbations, stating »Exacerbations in the past year«.

Point 5: ICS was not used in 11% of patients. What’s the reason? Why ICS was not prescribed in some asthma patients?

Response 5: Some patients had mild symptoms and were not taking ICS regularly or every day despite the recommendation. In this case, we did not include the patient as a subject with regular ICS therapy.

Minor comments

Point 6: Please add FEV1 (L), FVC (L), FVC (%), and FEV1/FVC (%).

Response 6: Unfortunately we were not able to obtain this information as FVC data are not available from the digital medical documentation of patients.

We have therefore upgraded the data of functional diagnostic with pre-and post-bronchodilator inhalation pulmonary function test (with data on FEV1 absolutely and in %).

Point 7: Please specify whether lung function is pre or postbronchodilator value.

Response 7: FEV1 in the original manuscript was pre-bronchodilatation. In the revised manuscript we have also added FEV1 post-bronchodilatation. For both measurements, we have absolute values in L, as well as percentages (%).

We have corrected in Table 1, Table 2, and in abbreviations list stating that FEV1 (%, L) preBD is defined as pre-bronchodilation forced expiratory volume in 1 s and FEV1 (%, L) postBD is defined as post-bronchodilation forced expiratory volume in 1 s.

Point 8: Control state and ACT score are missing. Please add.

Response 8: Unfortunately we were not able to obtain this information as ACT data are not available from the digital medical documentation of patients.

Point 9: How many patients are on ICSLABA? Please add.

Response 9: 29 patients were taking ICS/LABA. We have inserted this information in the footnote of Table 1 stating that »29 (81%) patients were taking ICS/LABA.«

Point 10: How many patients are on ICSLABALAMA? Please add.

Response 10: None of the included patients was taking LAMA.

Reviewer 4 Report

The main finding of this article authored by Matija Rijavec and entitled “T2-high asthma, classified by sputum mRNA expression of IL4, IL5, and IL13, is characterized by eosinophilia and severe phenotype” is to classify T2-high asthma based on mRNA expression of the T2 cytokines IL-4, IL-5 and IL-13 in sputum. The article is well written and easy to read, the methodology is rigorous and results are well interpreted. However, this is not really new as few papers exposed similar results, for example first authored by Peters MC as indeed mentioned and discussed by the current article. Nevertherless, I understand this information may be indeed very useful to discriminate T2-high and T2-low in the clinic. And it is very important that more than one laboratory confirm these results on different patients in order to really use these measurements internationally. Therefore, I deeply encourage the authors to reword the article, so that the reader can understand its usefulness, particularly what it brings more than previous articles of literature.

Specific questions:

What are the biological therapies of the patients included in this study? Please name them.

The authors state that “T2-high endotype is characterized by eosinophilia, and severe, difficult-to-treat asthma, which often requiring biological treatment to control exacerbations/symptoms”, but is it not because only biotherapies targeting Th2 cytokines are available?

Author Response

Response to Reviewer 4 Comments

The main finding of this article authored by Matija Rijavec and entitled “T2-high asthma, classified by sputum mRNA expression of IL4, IL5, and IL13, is characterized by eosinophilia and severe phenotype” is to classify T2-high asthma based on mRNA expression of the T2 cytokines IL-4, IL-5 and IL-13 in sputum. The article is well written and easy to read, the methodology is rigorous and results are well interpreted. However, this is not really new as few papers exposed similar results, for example first authored by Peters MC as indeed mentioned and discussed by the current article. Nevertherless, I understand this information may be indeed very useful to discriminate T2-high and T2-low in the clinic. And it is very important that more than one laboratory confirm these results on different patients in order to really use these measurements internationally. Therefore, I deeply encourage the authors to reword the article, so that the reader can understand its usefulness, particularly what it brings more than previous articles of literature.

Specific questions:

Point 1: What are the biological therapies of the patients included in this study? Please name them.

Response 1: Thank you for your careful examination of our manuscript. As stated in the footnotes of Tables 1 & 2 and in the Results section asthmatic patients included in the study needed biological treatment, either omalizumab or mepolizumab. We have also marked the patients in Figure 3 that needed biological therapy as recipients of omalizumab (â–¼) or mepolizumab (â–²).

Point 2: The authors state that “T2-high endotype is characterized by eosinophilia, and severe, difficult-to-treat asthma, which often requiring biological treatment to control exacerbations/symptoms”, but is it not because only biotherapies targeting Th2 cytokines are available?

Response 2: Thank you for this comment. We have made substantial changes and also highlighted in the Discussion section (lines 127-132) further highlighting what are the main novelties of our study »To our knowledge, this is the first report, which clinically adopted T2 sputum gene signatures and correlated this molecular profiling with the selection of adjunct biological therapy. Given that 94% of patients with the need for biological therapy demonstrated T2-high gene signature, sputum expression profiling should be considered clinically in all individuals with difficult-to-treat asthma.«

We have added in the Discission section (lines 201-212) a paragraph explaining the limitations and strengths of our study »There are some limitations of our study. First, the small number of included subjects, especially in the control group. Besides, a larger number of asthmatic patients would allow us to perform more subanalysis and we believe that some correlations would be even more evident. Another limitation of using gene expression profiling of selected cytokines (IL4, IL5, and IL13) in induced sputum for asthma stratification in comparison to unsupervised approaches employing several biomarkers is that prevalence of T2-high asthma might be underestimated. On the other hand, as described previously this approach is more robust, reproducible, and easier to implement into the clinic. Although we have shown that molecular profiling in induced sputum can be translated to clinical practice, the technique is complex and could only be performed in specialized and expert centers for specific cases in which other broadly used methods failed to classify patients as having T2-high or T2-low asthma.«

Round 2

Reviewer 2 Report

The authors have addressed properly all the suggestions made in the previous review report. In addition, as previously mentioned this is a very nicely written article and all conclusion are supported by the results.

However, from my point of view the novelty of the article remains on average, but you clearly pointed out this in the discussion section. Therefore, based on reviewer comments, it is on editor choice to evaluate if the novelty is sufficient to published this article in the present form.

I have only a minor comment:

Figure 3. You use unpaired t test. The correct test to use is One way ANOVA followed by Holm-Sidak multiple comparison test in case of normal distribution, or Kruskal Wallis followed by Dunn´s posh test as non parametric test.

Author Response

Response 1: Thank you again for your careful examination of our manuscript. We have used the correct statistical test and corrected Figure 3 accordingly.

Figure 3 legends »T2 gene mean calculated by combining the expression levels of IL4, IL5, and IL13 in controls and asthmatics divided by need for biological therapy either omalizumab (â–¼) or mepolizumab (â–²). The dotted line represents the T2 gene mean cut-off point for classification into T2-high/T2-low asthma. ANOVA test followed by Holm-Sidak multiple comparison test Unpaired t-test: * p < 0.05; *** p < 0.001; **** p < 0.0001.«

We have also added the description of the newly used statistical test in the Materials and Methods section / Statistical Analyses (lines 280-282) that now read »…or One way ANOVA test followed by Holm-Sidak multiple comparison test as appropriate.«

Reviewer 3 Report

Much more improvement can bee seen in the updated manuscript. I have no further comment.

Author Response

Response: We are thankful to reviewer for valuable suggestions and comments, which have helped us to improve the quality of our manuscript.
